# Mechanisms of Immune Evasion in Acute Lymphoblastic Leukemia

**DOI:** 10.3390/cancers13071536

**Published:** 2021-03-26

**Authors:** Agata Pastorczak, Krzysztof Domka, Klaudyna Fidyt, Martyna Poprzeczko, Malgorzata Firczuk

**Affiliations:** 1Department of Pediatrics, Oncology and Hematology, Medical University of Lodz, 91-738 Lodz, Poland; agata.pastorczak@umed.lodz.pl; 2Department of Immunology, Medical University of Warsaw, 02-097 Warsaw, Poland; krzysztof.domka@wum.edu.pl (K.D.); klaudyna.fidyt@wum.edu.pl (K.F.); mpoprzeczko@wum.edu.pl (M.P.); 3Doctoral School, Medical University of Warsaw, 02-091 Warsaw, Poland; 4Postgraduate School of Molecular Medicine, Medical University of Warsaw, 02-091 Warsaw, Poland

**Keywords:** acute lymphoblastic leukemia, immune system, bone marrow, microenvironment, immunotherapy, B cell, T cell, NK cell, macrophage, immune evasion

## Abstract

**Simple Summary:**

Studies conducted in a recent decade revealed that acute lymphoblastic leukemia cells exploit various mechanisms to avoid immune recognition and destruction by the immune system. As a consequence, leukemia cells become more resistant to treatment, which results in poor patient outcomes. In this review, we describe the interactions between acute lymphoblastic leukemia (ALL) cells and selected populations of immune cells within the bone marrow microenvironment that contribute to leukemia development, promotion, and relapse. We also discuss how various elements of the bone marrow niche affect ALL chemo- and immunotherapy and present recently discovered treatment strategies that restore or stimulate the immune response to ALL.

**Abstract:**

Acute lymphoblastic leukemia (ALL) results from a clonal expansion of abnormal lymphoid progenitors of B cell (BCP-ALL) or T cell (T-ALL) origin that invade bone marrow, peripheral blood, and extramedullary sites. Leukemic cells, apart from their oncogene-driven ability to proliferate and avoid differentiation, also change the phenotype and function of innate and adaptive immune cells, leading to escape from the immune surveillance. In this review, we provide an overview of the genetic heterogeneity and treatment of BCP- and T-ALL. We outline the interactions of leukemic cells in the bone marrow microenvironment, mainly with mesenchymal stem cells and immune cells. We describe the mechanisms by which ALL cells escape from immune recognition and elimination by the immune system. We focus on the alterations in ALL cells, such as overexpression of ligands for various inhibitory receptors, including anti-phagocytic receptors on macrophages, NK cell inhibitory receptors, as well as T cell immune checkpoints. In addition, we describe how developing leukemia shapes the bone marrow microenvironment and alters the function of immune cells. Finally, we emphasize that an immunosuppressive microenvironment can reduce the efficacy of chemo- and immunotherapy and provide examples of preclinical studies showing strategies for improving ALL treatment by targeting these immunosuppressive interactions.

## 1. Introduction

Acute lymphoblastic leukemia (ALL) results from a clonal expansion of abnormal, immature lymphoid progenitors of B cell or T cell origin that invade bone marrow (BM), peripheral blood, and extramedullary sites [1]. The incidence of ALL is highest among children aged 1–4 years and declines with age rising slightly after the age of 50. Several genetic alterations have been already found as drivers of ALL, uncovering the genetic heterogeneity of the disease. Age at diagnosis of ALL considerably affects the 5-year overall survival (OS), reaching 90% in pediatric patients and only 30–40% in adults older than 40 years [2]. High frequency of aggressive molecular ALL subtypes and an increased risk of treatment-induced toxicities contribute to poor prognosis in adult patients. ALL patients who do not respond to conventional chemotherapy are now treated with various types of immunotherapy. However, the efficacy of the immunotherapy is greatly affected by the interactions of leukemic cells with the microenvironment within the BM niche.

As other cancer cells, ALL cells, due to genetic alterations and dysregulated post-translational modifications, express neoantigens and can induce tumor-specific T cell responses. Indeed, though ALL is a malignancy with relatively low mutational burden, recent reports show that leukemic, antigen-specific T cells are present in the BM microenvironment [3]. There is also growing evidence that natural killer (NK) cells play a role in the immunosurveillance of ALL [4]. However, developing leukemia impairs the key components of the immune system responsible for mounting an anticancer response, particularly in patients poorly responding to treatment or at the relapse stage [5,6,7]. Cancer cells can avoid recognition and elimination by the immune system by various, cancer type-specific mechanisms, which are already well documented in solid tumors and are just being discovered in ALL. Downregulation or loss of human leukocyte antigen (HLA) class I molecules, which leads to impaired recognition of leukemic cells by cytotoxic T lymphocytes, is a rare but functionally significant mechanism of immune evasion in ALL [8]. Moreover, recent data uncovers the mechanism of T cell exhaustion operating in ALL [5,9,10], opening the possibility to employ appropriate immune checkpoint inhibitors to ALL treatment protocols. Finally, novel, high-throughput methods, in particular single cell RNA sequencing, reveal the important role of suppressive myeloid cell populations in promoting leukemia progression and hampering treatment efficacy [11].

In this review, we describe the crosstalk between ALL cells and the BM microenvironment that contributes to leukemia development, promotion, and relapse. We summarize recent findings on how ALL avoids destruction by the immune system and describe the interactions between ALL cells and selected populations of immune cells. The overview of the mutual interactions between ALL cells and the key cells of the BM microenvironment is presented in Figure 1.

We also discuss how various elements of the BM niche affect ALL chemo- and immunotherapy and present recently discovered treatment strategies that restore or stimulate the immune response to ALL.

## 2. Immunophenotype and Genotype of Acute Lymphoblastic Leukemia

Depending on the cell of origin, ALL is classified as a B cell precursor ALL (BCP-ALL) and T cell ALL (T-ALL). BCP-ALL accounts for 80–85% of ALL [2]. According to the Associazone Italiana Ematologia Oncologia Pediatrica and the Berlin-Frankfurt-Münster (AIEOP-BFM) Consensus Guidelines for Flow Cytometric Immunophenotyping, BCP-ALL is diagnosed when dominant leukemic clone in the BM shows strong expression of at least two of the following antigens: CD19, CD10, i(intracellular)CD22, or iCD79a [12]. Further classification of the blasts’ immunophenotype dissects four subtypes of BCP-ALL (BI: pro-B; BII: common; BIII: pre-B; and BIV: mature B), based on the differences in the expression of CD10, iIgM, λ, and κ chains [12]. BCP-ALL is also classified into several molecular subtypes according to the major (initiating) chromosomal abnormality and specific secondary lesions, which both result in a unique gene expression profile and chemosensitivity of leukemic cells [1]. This heterogeneous mutational landscape of BCP-ALL is reflected in the revised version of World Health Organization (WHO) classification of myeloid neoplasms and acute leukemias, which distinguishes eleven subtypes of ALL, based on the presence of somatic molecular lesions [13]. Genetic abnormalities that confer to BCP-ALL development may associate with the expression of the specific antigens in leukemic cells and serve as prognostic factors [12] (Table 1). Selected molecular lesions are currently used in the clinical practice to predict the risk for BCP-ALL recurrence in the individual patient and then adapt the intensity of the therapy accordingly. Moreover, dissection of genomically defined ALL subtypes led to the development of relevant molecularly targeted therapies that contributed to further significant improvement in patients’ survival. The examples are inhibitors targeting ABL1/ABL2, which increased the response rate among ALL patients with translocation t(9;22) (q34;q11), resulting in breakpoint cluster region–Abelson murine leukemia viral oncogene homolog 1 (*BCR–ABL1*) gene fusion (Philadelphia chromosome), or inhibitors of Janus kinase 2 (JAK2), which improved treatment outcome of individuals with Philadelphia chromosome-like ALL (Ph-like ALL) [14].

T-ALL represents approximately 12–15% and 20% of all ALL cases in children and adults, respectively [2]. It clearly shows a distinct biology, clinical features (hyperleukocytosis with extramedullary involvement of lymph nodes, central nervous system infiltration, and the presence of a mediastinal mass, arising from the thymus) and response to particular elements of therapy, as compared to BCP-ALL. The immunophenotype of T-ALL blast cells is distinguished by the presence of iCD3, iCD7, and weak expression or lack of iMPO [12]. There are four subtypes of T-ALL according to the European Group for the Immunological Classification of Leukemia (EGIL): pro-T EGIL T-I (iCD3^+^, CD7^+^), pre-T EGIL T-II (iCD3^+^, CD7^+^ and CD5/CD2^+^), cortical T EGIL T-III (iCD3^+^, Cd1a^+^, sCD3^+/−^), and mature-T EGIL T-IV (iCD3^+^, sCD3^+^, CD1a^−^) [25]. Early T cell Precursor (ETP) ALL represents an additional subtype, showing a unique immunophenotype (lack of expression of CD1a and CD8, weak CD5 expression, and expression of 1 or more myeloid or stem cell markers) and a specific genomic profile [26,27]. The genetic landscape of T-ALL varies according to the maturation stage of the dominant leukemic clone and includes chromosomal rearrangements resulting in gene fusions, point mutations, and copy number alterations (Table 2). Although these aberrations significantly contribute to malignant lymphoproliferation of T-ALL and several are even related to treatment outcome, none of them is presently used for risk stratification [28,29].

## 3. Treatment and Prognosis of Acute Lymphoblastic Leukemia

The first-line treatment of ALL usually consists of four phases: induction, consolidation, intensification, long-term maintenance, and directed therapy preventing central nervous system relapse of leukemia. In each phase, patients receive courses of different combinations of drugs, including glucocorticoids, vinca alkaloids (vincristine), anthracyclines (doxorubicin), antimetabolites (cytarabine, mercaptopurine, methotrexate), alkylating agents (cyclophosphamide), and L-asparaginase. While this intensive therapeutic approach leads to a 5-year OS of 90% in childhood ALL, adults achieve a 5-year OS of less than 45% [2]. Several studies have demonstrated improved long-term outcomes in adolescents and young adults (AYA) treated with pediatric-intensive chemotherapy [30]. Additionally, in case of chemotherapy failure or insufficiency, ALL patients may be treated with various forms of immunotherapy (Table 3), including the oldest one-hematopoietic stem cell transplantation (HSCT). The curative effect of HSCT results from direct cytotoxicity from the chemo-radiotherapy administered in the conditioning regimen, along with an immune effect termed graft-versus-leukemia (GVL) [31].

HSCT is reserved for patients with high risk ALL who show specific genetic abnormalities (e.g., low hypodiploidy) and/or persistent minimal residual disease (MRD), are fit, and have an available donor [2]. Whereas HSCT is still perceived as a standard consolidative therapy preventing relapse in many adult patients with Ph-negative ALL [32], the role of HSCT in childhood ALL is continuously redefined as advances in immunotherapy and targeted treatment are made. A detailed discussion of the indications of HSCT in childhood ALL, as depending on the molecular aberrations and response to therapy, was presented by Merli et al. [33].

Bone marrow transplantation has severe limitations, e.g., graft versus host disease (GVHD), increased infection rates due to delayed immune reconstitution, and difficulties in finding suitable, HLA-matched donors. For BCP-ALL patients, other forms of more selective immunotherapy targeting B cell-specific antigens, CD19, CD20, and CD22, are already available [2,34]. The first group includes therapeutic monoclonal antibodies (mAbs) and their derivatives, such as rituximab (anti-CD20 “naked” mAb), inotuzumab ozogamycin (anti-CD22 mAb conjugated with a drug calicheamicin), and blinatumomab (anti-CD19/anti-CD3 bispecific T cell engager) [35,36,37]. Since 2017, cellular immunotherapy using patient-derived T cells, genetically modified with chimeric antigen receptors (CAR-T cells), has been approved. CARs are synthetic constructs that consist of an extracellular antibody-derived domain that is responsible for antigen recognition, intracellular activating domain derived from T cell receptor (CD3ζ chain), and different co-stimulatory domains (CD28, 4-1BB, ICOS, OX40). Upon introduction into patient’s T cells, CARs reprogram the cells to recognize antigens present on leukemic cells, which results in T cell activation and leukemic cell death [38]. Tisagenlecleucel (autologous, anti-CD19 CAR-T cells) is approved by the FDA (U.S. Food and Drug Administration) for the treatment of relapsed or refractory pediatric and adult BCP-ALL. Despite a spectacular initial response, the long-term outcome of refractory/relapsed BCP-ALL patients treated with CAR-T cells is limited, with overall survival reaching only 12.9 months [39]. The development of CAR-T cells against other B cell antigens is ongoing. Various types of immunotherapy, already approved and tested in ALL patients, are presented in Table 3.

**Table 3 cancers-13-01536-t003:** Overview of immunotherapy used against acute lymphoblastic leukemia.

Applied in clinics (BCP-ALL)			
Type of Immunotherapy	Drug/Therapy Name	Mechanism of Action	References/ Clinical Trial No.
HSCT		Infusion of hematopoietic stem/progenitor cells	[40]
mAbs	Blinatumomab	Anti-CD19/CD3 bi-specific T-cell-mAb, which binds simultaneously to CD3-positive cytotoxic T cells and to CD19-positive B cells; endogenous T cells recognize and eliminate CD19-positive ALL blasts	[41] NCT02013167 NCT03628053
	Rituximab	Humanized murine mAb targeting CD20 on BCP-ALL cells	[42]
	Ofatumumab	Second-generation anti-CD20 mAb. Binds to a different epitope on the CD20 than Rituximab.	[43]
ADC (antibody-drug conjugates)	Inotuzumab ozogamycin	Humanized anti-CD22 antibody conjugated to a calicheamicin (cytotoxic drug); the CD22–conjugated complex is rapidly internalized and calicheamicin is released, which induces DNA double strand breaks	[44]
CAR-T cells	Tisagenlecleucel	Chimeric antigen receptor (CAR) T-cells targeting CD19 antigen and containing 4-1BB zeta co-stimulatory domain	[45] NCT03123939
**Tested in clinical trials (BCP-ALL)**			
mAbs	Epratuzumab	Humanized anti-CD22 mAb (IgG1)	[46] NCT02844530 NCT01354457 NCT01802814
	Alemtuzumab	Fully humanized anti-CD52 mAb	[47,48]
	Blinatumomab + Nivolumab +/− Ipilimumab	Anti-CD19/CD3 bi-specific T-cell-mAb + Humanized anti-PD-1 mAb +/− Humanized anti-CTLA4 mAb	NCT02879695
	Blinatumomab +Nivolumab	Anti-CD19/CD3 bi-specific T-cell-mAb + Humanized anti-PD-1 mAb	NCT04546399
	Blinatumomab + Pembrolizumab	Humanized anti-PD-1 mAb	NCT03160079
	TTI-621 + Rituximab/Nivolumab	TTI-621 (SIRPαFc) is a soluble recombinant fusion protein composed of the N-terminal CD47 binding domain of human SIRPα and the Fc domain of human immunoglobulin (IgG1); TTI-621 binds to CD47 and prevents “do not eat” (anti- phagocytic) signaling	NCT02663518
ADC	Denintuzumab	Humanized anti-CD19 antibody conjugated to a microtubule-disrupting agent monomethyl auristatin F (MMAF)	[49]
	Coltuximab ravtansine (SAR3419)	Anti-CD19 monoclonal antibody conjugated to potent inhibitor of tubulin polymerization and microtubule assembly, maytansinoid, DM4	[50]
CAR-T cells	CD19-CD22 CAR-T	Modified autologous T cells expressing anti-CD19 and anti-CD22 CARs	[39] NCT04626765
	CD22 CAR-T	Modified autologous T cells expressing anti-CD22 CARs	[39] NCT04626765
	CD19–28z CAR-T	Modified autologous T cells expressing anti-CD19 CARs with CD28 co-stimulatory domain	[39]
NK cells	Allogenic activated NK cells	Infusion of IL-15/IL-21-activated NK cells after HLA-mismatched HSCT	[51]
	Allogenic activated NK cells	Activated and expanded natural killer cells (NKAEs) from haploidentical donor infused to patients	NCT02074657
	Autologous NK cells	Enriched and expanded autologous NK cells	NCT02185781
	Cord blood NK cells	Personalized cord blood (CB)-derived NK cells for HLA-C2/C2 patients after chemotherapy	NCT02727803
	Cord blood NK cells	CD19-CD28-zeta-2A-iCasp9-IL15-transduced cord blood natural killer (CB-NK) cells recognizing CD19^+^ tumor cells	NCT03056339
**Tested in clinical trials (T-ALL)**			
mAbs	Isatuximab	Anti-CD38 mAb	NCT02999633 NCT03860844
	Daratumumab	Anti-CD38 mAb	NCT03384654
	Alemtuzumab	Anti-CD52 mAb	NCT00199030 NCT00061048 NCT00061945
CAR-T cells	CD4 CAR-T	Modified T cells expressing anti-CD4 CARs	NCT03829540 NCT04162340
	CD5 CAR-T	Modified T cells expressing anti-CD5 CARs	NCT03081910
	CD7 CAR-T	Modified T cells expressing anti-CD7 CARs	NCT04004637 NCT04264078 NCT03690011 NCT04033302 NCT04480788
NK cells	CD7 CAR NK cells	Modified NK cells expressing anti-CD7 CARs	NCT02742727

Abbreviations: ADC, antibody-drug conjugates; Casp9, caspase 9; CTLA-4, cytotoxic T-lymphocyte-associated protein 4; HSCT, hematopoietic stem-cell transplantation; mAbs, monoclonal antibodies; PD-1, programmed death receptor 1; SIRPαFc, signal regulatory protein α fragment crystallizable.

In contrast to the wide range of immune targets in BCP-ALL, treatments targeting T-ALL-specific antigens are less available in clinical practice. Nevertheless, Hening et al. has recently shown an effective eradication of residual disease and prolonged disease-free survival in patients with advanced relapse of T-ALL who obtained recombinant monoclonal anti-CD38 antibody (daratumumab) [52]. In addition, mimicking negative selection of T lymphocytes in the thymus using anti-CD3 antibody showed promise in preclinical studies in CD3/TCR-positive T-ALL [53].

There is a wide range of biological and clinical factors predicting the outcome of ALL, including age, sex, race, blood count at diagnosis, immunophenotype, and genetic alterations, but the key prognostic determinant is the response to treatment measured through serial minimal residual disease (MRD) monitoring [2]. Chemosensitivity of ALL, as well as response to immunotherapies, depend on intrinsic biological features of cancer cells but are also modified by diverse interactions between leukemic cells and the BM microenvironment. The long-term response to immune-based therapy might be limited as a result of clonal evolution of leukemia. This process promotes selection of ALL subclones exhibiting treatment-refractory phenotype through progressive and diverse accumulation of genetic alterations by neoplastic cells under therapeutic pressure [54,55]. There are several examples proving that clonal evolution and intratumor heterogeneity affect the efficacy of targeted immunotherapy [56,57]. One of the most illustrative in this context is successful elimination of CD19^+^ cells by patients with *KMT2A*-rearranged B-ALL receiving CD19 CAR-T [54].

## 4. Bone Marrow Niche as a Sanctuary Site Supporting Acute Lymphoblastic Leukemia Development

Physiologically, BM niche is a specialized area of interactions between cells, matrix, biophysical forces, and metabolites. These factors provide the appropriate conditions for hematopoietic stem cells (HSCs) quiescence, self-renewal, proliferation, multi-lineage differentiation capacity, and localization [58]. There are two functionally distinct BM niches: osteoblastic (endosteal) and vascular. The osteoblastic niche is located at the endosteum and contains diverse non-hematopoietic cell types, including osteoblasts, osteoclasts, glial non-myelinating Schwann cells, and regulatory T cells (Tregs). The vascular niche is enriched at the sinusoidal walls and comprises mesenchymal stem cells (MSCs), reticular cells, leptin receptor-positive perivascular stromal cells, macrophages, arteriolar and sinusoidal endothelial cells, as well as other immune cells. All these diverse cell types provide juxtacrine and paracrine signals (growth factors, chemokines, cytokines, morphogens, and adhesion molecules) to either normal hematopoietic or leukemic cells [59,60]. Osteoblastic niche supports HSCs quiescence while vascular niche stimulates cell proliferation, differentiation, and mobilization [61]. Additionally, multi-lineage differentiation of HSCs, as well as progression of leukemia within the niche, are regulated by extracellular matrix proteins (heparin sulfate, tenascin-C, and osteopontin) and by the extrinsic factors (calcium ions, the oxygen tension, the local concentration of reactive oxygen species (ROS), nitric oxide (NO), and pH) [59].

### 4.1. Interactions of Leukemic Cells with the Bone Marrow Microenvironment

Leukemogenesis can no longer be perceived as a process that exclusively affects HSC or more differentiated lineage precursors. Some genetic alterations, epigenetic changes, and transcriptomic variations are already present within stromal compartment at cancer diagnosis, but their incidence in patients with ALL has not been precisely defined [62,63,64]. Moreover, the possible role of the microenvironment in promoting hematopoietic dysregulation is not restricted to the small percentage of patients harboring germline predisposition to leukemia. At least, in the case of myeloid malignancies, genomic and transcriptomic variations in stromal compartment have been also observed outside of constitutional defects [65]. Moreover, leukemic cells modify the BM microenvironment in a way that allows maintenance of cancer cells survival and the protection against chemotherapy. Blasts may affect the osteoblastic niche, leading to the loss of osteoblastic cells and an impaired function of normal HSCs [60]. They contribute to the remodeling of the extracellular matrix (ECM) through secretion of matrix metalloproteinase-9 (MMP-9) [66]. Interestingly, overexpression of C-X-C chemokine receptor type 4 (CXCR4) on leukemic cells is associated with extra-medullary organs involvement and poor patients’ outcome [67]. Due to the strong effect of CXCR4-mediated interaction between the BM microenvironment and leukemic cells, several CXCR4 antagonists have been successfully tested in preclinical models. They decreased the number of leukemic cells in peripheral blood and limited the dissemination of ALL cells to extra-medullary sites [68]. However, these promising biological effects did not fully translate into clinical practice. Plerixafor, a reversible CXCR4 antagonist, showed only a modest therapeutical effect among heavily pretreated pediatric patients diagnosed with refractory/relapsed acute leukemias [69].

An interaction between leukemic cells and the BM microenvironment may also be directly associated with genetic alterations of leukemic cells and might be targeted to augment the existing treatment strategies. This can be illustrated by genetic abnormalities within the *IKZF1* gene in kinase-driven (Ph-positive and Ph-like) ALL, which result in acquisition of a hematopoietic stem cell-like phenotype and elevated cell surface expression of adhesion molecules. The acquisition of this phenotype causes extravascular invasion, residence in the niche, and activation of integrin signaling pathways. This leads to peculiar BM stromal cell adhesion and mislocalization of the *IKZF1*-mutated ALL cells and results in decreased chemosensitivity. Interestingly, retinoids and focal adhesion kinase (FAK) inhibitors restore sensitivity to chemotherapy [70].

### 4.2. Growth-Supporting and Protective Role of Mesenchymal Stem Cells against Leukemic Cells

MSCs are essential components of the hematopoietic microenvironment. They are mainly located close to the vascular BM niche, where they give rise to different cell lineages, including osteoblasts, chondrocytes, adipocytes, and myocytes. MSCs produce HSC-supporting substances, including chemokines (stromal cell-derived factor 1(SDF1/CXCL12)), cytokines, angiopoietin, stem cell factor (SCF/Kit ligand), galectin-3, extracellular matrix (ECM) proteins, and metabolites. Examples of the effects of these factors on leukemia development, progression, and modulation of response to therapy are discussed below [60,61]. Importantly, human MSCs co-cultured ex vivo with primary ALL cells support ALL cells’ survival and enable in vitro drug testing that allows to better predict in vivo response [69,71,72].

ALL cells tend to engraft in a vascular niche that is particularly enriched in E-selectin and stromal-derived factor 1 (SDF-1/CXCL12). Together with IL-7, CXCL12 stimulates proliferation of both T- and BCP-ALL [73,74]. Other cytokines (IL-1, IFN-γ, and TNF-α) and the soluble HLA-G in the bone marrow microenvironment may also contribute to T-ALL recurrence by inducing immunotolerance [66,75]. Recently, tunneling nanotubes were identified as a mechanism of the crosstalk between ALL cells and MSCs within the niche. This intercellular communication leads to the secretion of prosurvival cytokines (IL-8) and chemoattractants interferon gamma-induced protein/ CXC chemokine ligand 10 (IP10/CXCL10), monocyte chemotactic protein-1 (MCP-1)/CCL2) enhancing the resistance of leukemic cells to steroids [76]. In addition, via the expression of heme oxygenase-1 and the release of vascular endothelial growth factor (VEGF), MSCs promote resistance to vincristine [77]. MSCs can also protect ALL cells against the treatment by secretion of soluble or stroma-cell-bound galectin-3, which stimulates expression of endogenous *LGALS3* mRNA and activates the tonic NF-κB pathway in leukemic cells [78]. The proliferation and dissemination of ALL cells in the niche is also stimulated by MSC-derived ECM proteins, such as periostin and osteopontin. Periostin induces the expression of C-C Motif Chemokine Ligand 2 (CCL2) in BCP-ALL cells [79]. Osteopontin controls ALL cell dormancy in the niche and its neutralization sensitizes ALL cells to chemotherapy with cytarabine [80]. Finally, MSCs also secrete metabolites, such as asparagine, that reduces the cytotoxicity of L-asparaginase [81].

### 4.3. Leukemia-Induced Loss of Immunosurveillance within the Bone Marrow Niche

Bone marrow provides a microenvironmental framework supporting development and function of all immune cells and acts as a host for various mature cell types, including B and T cells, plasma cells, dendritic cells, neutrophils, and macrophages that reside in the BM niche [61]. Therefore, the niche becomes the main and essential place of mutual interactions of cancer and immune cells and, as a result of these interactions, leukemic cells should be recognized and eliminated. However, during leukemogenesis, proliferating blasts affect the differentiation and function of immune cells, leading to loss of immunosurveillance and cancer progression. Below we provide examples of how developing leukemia affects key cells of the immune system involved in the immunosurveillance, such as myeloid cells, NK cells, and various subpopulations of T cells.

## 5. Myeloid Cells

Cells of myeloid origin at various stages of differentiation and maturation contribute to immune evasion in ALL. The immature myeloid populations involved in immunoregulation are collectively referred to as myeloid-derived suppressor cells (MDSCs). The mature myeloid populations include granulocytes, monocytes, macrophages, and dendritic cells (DCs). All myeloid cell types have distinct immunological functions and can be identified by their immunophenotype as described in [82,83,84,85].

The development of myeloid lineage begins in the bone marrow from granulocyte-monocyte progenitors (GMPs) or monocyte-dendritic cell progenitors (MDPs) [86]. Some of them do not undergo a full differentiation into monocytes and remain in an immature state to become MDSC - immunoregulatory cells, best known for their role in promotion of tumor development, also in hematological malignancies [87,88]. Two distinct populations of MDSCs, granulocytic (G) and monocytic (M), can be distinguished depending on the phenotypic differences, as reviewed in [89,90]. This classification also implies functional differences: G-MDSCs induce immunosuppression mostly by reactive oxygen species (ROS)-dependent mechanisms, while M-MDSCs act rather by secretion of immunosuppressive cytokines (IL-10, TGFβ), arginase, and NO.

Unlike MDSCs, most myeloid progenitors continue their development to become granulocytes or monocytes. The latter ones can be divided into three main subsets depending on the expression of CD14 and CD16 surface markers. Around 90% of monocytes are characterized by high expression of CD14 and are known as classical (CD14^++^CD16^−^) monocytes. The other 10% are non-classical (CD14^+^CD16^++^) and intermediate (CD14^+^CD16^+^) monocytes [91]. Monocytes act as early responders to inflammation or infection. Additionally, non-classical monocytes are involved in tissue repair and healing processes [92].

After several days of circulation in the blood, monocytes tend to migrate to other tissues and transform into macrophages or dendritic cells. Both populations are involved in phagocytosis and antigen presentation. Macrophages are traditionally divided into two subtypes: classically activated, antitumor M1 macrophages, and alternatively activated, tumor-supporting M2 macrophages [93]. This paradigm is now viewed as a basis for a spectrum of numerous macrophage phenotypes located between those two extremes [94,95,96]. Macrophages display great phenotypic flexibility, depending on microenvironmental stimuli. The studies on different macrophage subtypes revealed that in the tumor microenvironment, some of the macrophages acquire immunosuppressive properties and become tumor associated macrophages (TAM), or leukemia associated macrophages (LAM). Such macrophages promote tumor cells proliferation, migration, and angiogenesis [97]. Most LAMs are of the M2-like phenotype. Similarly, the dendritic cells are also divided into several subpopulations with distinct immunological features [98].

### 5.1. Myeloid-Derived Suppressor Cells

Though the pro-tumorigenic function of MDSCs is well established in solid tumors and some hematological neoplasms, their role in the development and progression of ALL is still poorly understood. Several studies have reported that the numbers of MDSCs are higher in the blood and the BM of patients diagnosed with BCP-ALL in comparison with healthy donors [10,99]. It was also shown that some chemotherapeutic drugs used in the treatment of ALL, as cyclophosphamide and doxorubicin, increase the already elevated number of circulating MDSCs [99,100].

Substantial evidence of G-MDSCs (but not M-MDSCs) being involved in evading immune surveillance in BCP-ALL was documented in the study of Liu et al. [101]. The authors compared the numbers of G-MDSCs in peripheral blood and BM of BCP-ALL patients and healthy donors and discovered a significant elevation of these suppressive cells in patients’ samples. Moreover, the G-MDSCs numbers positively correlated with blast clearance and a worse therapeutic response. Ex vivo tests confirmed that G-MDSCs suppress T cell and NK cell responses via ROS [101].

### 5.2. Non-Classical Monocytes

The accumulation of non-classical monocytes was the major immune cell alteration observed recently in BCP-ALL patients’ BM [11]. To establish the main differences between a healthy and leukemic BM microenvironment, the authors employed a single-cell RNA sequencing-based approach. They found that the myeloid cell compartment was most significantly changed in BCP-ALL in patients’ BM and that leukemic cells promoted non-classical monocyte differentiation, likely in response to vascular damage. Accordingly, the pan-myeloid depletion via colony stimulating factor 1 receptor (CSF1R) blockade increased sensitivity to nilotinib in an in vivo mouse model of BCP-ALL, indirectly confirming the role of non-classical monocytes in the protection of leukemic cells [11].

### 5.3. Macrophages and Dendritic Cells

Elevated levels of macrophages expressing M2 markers, including CD68, CD163, and CD206, were found in the BM of BCP-ALL patients [10,102]. Valencia et al. reported that culturing of macrophages in ALL-conditioned medium resulted in the upregulation of tumor-promoting cytokines (IL-10, TGF-β), enzymes (IDO1, MMP9), and VEGF. Similarly, human dendritic cells have also acquired the same immunosuppressive features after analogous culturing [103].

M2 macrophages were also shown to provide pro-survival signals to T-ALL cells. Human macrophages cultured in vitro secreted insulin-like growth factor IGF1, which contributed to increased survival of T-ALL cells expressing IGF1-R (insulin-like growth factor 1 receptor) [104]. Accordingly, depletion of myeloid cells led to reduced leukemic burden and improved survival of mice in the LN3 transgenic model of T-ALL [104].

Apart from promoting M2 differentiation, ALL cells can also overcome immune surveillance by escaping the effector mechanisms of functional macrophages. This can be done by overexpression of CD47 - a surface protein that acts as a “do not eat me signal” for macrophages, by triggering its inhibitory anti-phagocytic pathway mediated by the SIRPα receptor. Chao et al. have confirmed the presence of CD47 on T-ALL and BCP-ALL cells and used the humanized anti-CD47 antibody to promote ALL cells’ phagocytosis in vitro and inhibit ALL engraftment in NSG mice [105]. Accordingly, a fully humanized anti-CD47 antibody induced an efficient phagocytosis of human T-ALL cell line CCRF in vitro, as well as inhibited engraftment of this cell line in immunodeficient BALB/c nude mice. A strategy aiming to prevent this inhibitory pathway, employing recombinant fusion protein composed of a domain of SIRPα and Fc of human IgG1, is currently being tested in a clinical trial (NCT02663518).

### 5.4. Use of Myeloid Cells in Acute Lymphoblastic Leukemia Therapy

An interesting approach combining monocyte-based adoptive immunotherapy and gene therapy for BCP-ALL treatment was presented by Escobar et al. [106]. The high infiltration of BM microenvironment by monocytes was utilized to counteract immune evasion. An innovative gene therapy was designed to deliver IFNα into the bone marrow niche by using genetically engineered monocyte progenitors. Such an approach, tested in a mouse model of BCP-ALL, resulted in the promotion of T cell activity and synergized with tumor-specific CAR-T cells treatment, successfully overcoming inhibitory signals from the leukemic microenvironment [106].

## 6. Natural Killer Cells

Natural killer (NK) cells are innate lymphoid cells that recognize and kill virus-infected or malignant cells (target cells). They utilize several pathways to kill target cells, which are extensively summarized in [107,108]. The ability of NK cells to kill ALL blasts depends on the balance between the activating and inhibitory receptors on NK cells as well as by the presence of their corresponding ligands on ALL cells [109]. The inhibitory receptors, among others, include killer immunoglobulin-like receptors (KIR), such as killer cell immunoglobulin-like receptor 2DL (KIR2DL), killer cell immunoglobulin-like receptor 3DL (KIR3DL), and natural killer group 2A (NKG2A). These receptors recognize classical and non-classical HLA class I molecules. The activating receptors include killer cell immunoglobulin-like receptor 2DS (KIR2DS) and killer cell immunoglobulin-like receptor 3DS (KIR3DS); natural killer group 2D (NKG2D); DNAX accessory molecule 1 (DNAM-1); and Natural Cytotoxicity Receptors (NCRs), e.g., NKp46. The NKG2D receptor recognizes the stress inducible molecules, MHC class I chain-related proteins A and B (MIC A/B), and UL16-binding proteins (ULBP), whereas DNAM-1 binds poliovirus receptor (PVR) and Nectin-2. The ligands for the other activating receptors are not well defined. The occurrence of HLA molecules on normal cells protects them from killing by NK cells [107,110,111,112]. As the genes encoding HLA molecules are polygenic and polymorphic, it was shown that specific alleles may promote leukemia development, highlighting the role for NK cells in ALL immunosurveillance. The occurrence of the C2 variant of HLA-C, which binds the inhibitory KIR2DL1 receptor with high affinity [113], correlated with leukemia susceptibility and risk of late relapse [114].

### 6.1. Expression of Ligands for NK Receptors on Leukemic Cells

Several studies reported that ALL blasts are not effectively killed by NK cells in vitro [115,116,117,118]. Although the reasons are not yet precisely understood, it is known that developing leukemia modifies NK cells’ education and maturation. Moreover, leukemic blasts avoid NK cell-mediated killing by the expression of selected ligands for NK cell receptors. Romanski et al. showed that BCP-ALL cell lines and primary cells, but not T-ALL cells, are resistant to NK cell (NK-92)-mediated lysis due to low levels of MICA/B ligands for activating receptor NKG2D [115,118]. Interestingly, adult and pediatric ALL blasts showed different expression of the ligands for the NKG2D and DNAM-1 receptors [119]. Higher levels of Nec-2 (ligand of DNAM-1), ULBP1, and ULBP3 (ligands of NKG2D) were observed in pediatric patients as compared to adults. Moreover, differences in the expression of the ligands for NK cell receptors were observed between genetically defined BCP-ALL subgroups. Blasts from patients carrying the *BCR-ABL1* gene fusion had significantly higher levels of NKG2D and DNAM-1 ligands and were more sensitive to NK cell-mediated lysis than blasts without this genetic aberration [119]. Although lower levels of HLA-C and HLA-E, the ligands for NK cell inhibitory receptors, were also observed in BCP-ALL primary cells collected at diagnosis, the activation of NK cells due to “missing-self” was not observed [116]. All the above results suggest that ALL blasts escape from NK cell lysis predominantly by downregulation of the ligands for NK cell-activating receptors.

### 6.2. Alterations in the Numbers and Activity of NK Cells

Patients with ALL show differences in the number, phenotype, and activity of NK cells. Early studies showed that patients suffering from ALL have abnormal percentages and absolute numbers of NK cells [120]. Moreover, Valenzuela-Vazquez et al. confirmed these observations in Hispanic ALL patients, especially in those diagnosed with T-ALL [121]. Already in 1989, Gabrilovac et al. observed impairment of NK cell function in ALL [122]. More recent studies confirmed alterations in the phenotype and cytotoxicity of NK cells derived from ALL patients [118,121]. As compared to healthy donors, NK cells from pediatric ALL patients had lower levels of NKp46 activating receptors and higher levels of inhibitory NKG2A receptors [118]. Moreover, factors limiting NK cell activity (e.g., low expression of activating receptor NKp46) correlated with MRD positivity [123]. Rouce et al. observed that the dysfunction of NK cells from ALL patients results from the release of TGFβ1 by BCP-ALL cells, which leads to the activation of the TGFβ/SMAD pathway and in consequence to NK cell dysfunction [118].

### 6.3. NK Cells in Acute Lymphoblastic Leukemia Immunotherapy

NK cells also greatly affect the efficacy of ALL therapy. Patients’ NK cells are the major cytotoxic effector cells that mediate the therapeutic antibody-dependent cellular cytotoxicity (ADCC) effect. The NK cell receptor responsible for ADCC is CD16, which recognizes the Fc fragment of immunoglobulin G (FcγRIII). NK cells are particularly important for the efficacy of “naked” therapeutic monoclonal antibodies, such as rituximab. Bispecific antibodies triggering NK cells, named bispecific killer cell engagers (BiKE), with analogy to BiTE, were effective against primary ALL cells in preclinical tests [124]. It is worth emphasizing, however, that the effectiveness of such therapeutic monoclonal antibodies, both mono and multi-specific, is determined by the condition and cytotoxic capability of the patient’s NK cells, which is compromised. In this context, immunotherapy using NK cells derived from healthy donors may be a more effective therapeutic option.

Extensive work is underway to optimize in vitro NK cell expansion protocols. The activation of healthy donor-derived NK cells in vitro by interleukin 2 (IL-2), interleukin 12 (IL-12), interleukin 15 (IL-15), and interleukin 18 (IL-18) results in more effective lysis of ALL blasts [125,126,127]. Recently, Liu et al. established a novel platform for selective expansion of natural killer group 2 C (NKG2C)^+^ NK cells that express a single KIR receptor (KIR2DL3 or KIR2DL1) and specifically kill allogeneic ALL cells lacking the cognate HLA ligand [127]. To enhance the killing potential and increase cancer specific recognition, NK cells may be genetically modified to express CAR molecules. A CD19-specific CAR-modified NK cells reveal high cytotoxic activity against malignancies derived from mature B cells (chronic lymphocytic leukemia and non-Hodgkin lymphoma), without causing toxic effects [128]. CD19-CAR-NK cells also presented preliminary efficacy against BCP-ALL cell lines in in vitro assays [129]. NK cell-based immunotherapies tested against ALL are presented in Table 3.

Finally, NK cells also play a central role in HSCT. The HSCT from HLA-haploidentical donors has been claimed to benefit from the graft-vs-leukemia (GVL) effect, which is achieved by alloreactive NK cells from the donor [130,131,132]. The molecular basis for the NK cells’ alloreactivity is the recognition of specific ligands for NK cell activating receptors and lack of recognition by the inhibitory receptors (“missing-self” reactivity) [64]. Research from the last decade provides information how HLA and KIR genotypes of the donors shape NK cell repertoire and alloreactivity of NK cells [117]. Pende et al. presented that the activating receptor KIR2DS1, which recognizes HLA-C2 alleles on ALL cells, plays a major role in the lysis of leukemia cells by donor NK cells [130]. These results may guide optimal selection of HSCT donors for a more effective treatment of ALL.

## 7. T Cells

T cells possess a wide range of effector functions and play a substantial role in adaptive immune responses. Based on the expression of either CD4 or CD8 molecules, T cells are respectively classified as T helper (Th) or T cytotoxic cells (Tc). Physiological function of Tc cells is to eliminate host cells displaying foreign antigens. Several subsets of Th cells on the other hand, are responsible for the regulation of cell-mediated and humoral immune responses. A specific subset of CD4^+^ T cells expressing CD25 antigen and intracellular transcription factor Forkhead box P3 (FOXP3), known as regulatory T cells (Tregs), inhibits adaptive immune responses mainly through interaction with other subpopulations of T cells and antigen-presenting cells [133].

It is currently well established that T cells are capable of recognizing and eradicating tumor cells. Accordingly, exploring the landscape of tumor-specific and tumor-associated antigens and their implication for immunotherapies in ALL has been a goal of several preclinical studies [134,135,136,137]. Deeper insight into the occurrence of ALL antigens-specific CD8^+^ T cells was provided in a recent study by Zamora et al. [3]. Employing high-throughput approaches, the authors identified cancer specific antigens, which elicited T cells expansion and efficient responses to majority of tested neoantigens, including ETV6-RUNX1 fusion protein. Nevertheless, the sole occurrence and presentation of antigens with a proven immunogenic potential is insufficient to persistently activate T cells and provide immune responses that ensure tumor clearance. Two main mechanisms that attenuate the T cell-mediated antitumor immune response are well recognized: the expression of inhibitory checkpoint receptors on conventional T cells, and the accumulation and augmented suppressive function of Tregs. Moreover, in the case of T-ALL, a T cell-derived malignancy, even more challenges for potent T cell-mediated immune responses/immunotherapies are met [138]. This is partially due to the target antigens overlapping between leukemic and normal T cells, which can lead to various off-target effects.

### 7.1. Immune Checkpoints and Their Ligands

Among others, inhibitory receptors present on T cells include cytotoxic T cell antigen 4 (CTLA-4), programmed cell death protein 1 (PD-1), T cell immunoglobulin and mucin domain-containing protein 3 (TIM-3), and lymphocyte-activation gene 3 (LAG-3). The detailed characteristics of the various inhibitory pathways and their involvement in T cell exhaustion (defects in T cell effector functions) are extensively reviewed in [139,140,141,142]. Therefore, to ensure easier interpretation of results discussed in this review, we only briefly summarized several components of the immune checkpoint pathways, and their main mechanisms of action (Table 4). Interactions between inhibitory molecules on T cells and their respective ligands on the leukemic cells/antigen-presenting cells investigated so far in ALL patients and in ALL murine models, include CTLA-4/CD80 or CD86, PD-1/PD-L1, TIM-3/galectin-9, TIGIT/CD155, and CD200/CD200R.

CTLA-4 is the first immune checkpoint molecule described and the first clinical target for immune checkpoint inhibition [143,144]. Using multiplexed immunohistochemistry, Hohtari et al. [10] compared healthy and leukemic bone marrow derived from adult BCP-ALL patients and revealed high CTLA-4 expression in CD4^+^ and CD8^+^ T cells in patients’ bone marrow. Increased CTLA-4 expression was also confirmed in various subsets of peripheral T cells in pediatric BCP-ALL. Interestingly, the levels of this checkpoint molecule were more pronounced in high risk ALL subtypes and correlated with decreased relapse-free survival [7].

Other inhibitory interactions involving leukemic T cells, such as PD-1/PD-L1, TIM-3/galectin-9, TIGIT/CD155, and their effects on the immune anti-leukemic responses were recently described [3,5,158]. For instance, in a syngeneic murine BCP-ALL leukemia model, the PD-1 level was increased in CD4^+^ and CD8^+^ T cells, and, to some extent, ALL-induced PD-1 expression was independent of TCR activation [159]. Interestingly, PD-1 expression alone was not responsible for T cell exhaustion in this BCP-ALL murine model as the immune checkpoint inhibition had no impact on disease progression. In addition, the authors presented that introducing murine CD19 CARs into T cells isolated from leukemia-bearing mice did not fully improve their effector functions, indicating significant exhaustion of the leukemic T cells. In contrast to this finding, only small amounts of neoantigen-specific CD8^+^ T cells expressed PD-1 in human ALL [3]. Hence, it seems that despite ambiguous PD-1 expression, the PD-1/PD-L1 pathway has little impact on T cell responses in ALL, and that other inhibitory pathways may be involved in this process. Of note, Anand et al. [9] presented the involvement of TIM-3 and its ligand, galectin-9, in the inhibition of CD8^+^ T cell responses in early T-cell precursor ALL. Moreover, higher numbers of TIM-3^+^/PD-1^+^ CD4^+^ T cells predicted poor treatment outcome in adult BCP-ALL [10]. In line with this data, Blaeschke et al. [5] presented that TIM-3^+^ CD4^+^ T cells were enriched in BCP-ALL pediatric patients and were strongly associated with increased risk of relapse. Genetically modified, TIM-3 overexpressing T cells co-cultured in vitro with ALL cells possessed lower activation and proliferation rates than observed in wild-type T cells. Interestingly, the authors discovered that ALL-mediated induction of TIM-3 on T cells is mediated by CD200, yet another immune checkpoint molecule; however, the precise mechanism how CD200 augments TIM-3 expression needs further evaluation.

Inhibition of T cell responses elicited by interaction of CD200 with its cognate receptor, CD200R, was confirmed in various hematological cancers [160,161], showing potential for development of novel checkpoint inhibitors. Importantly, CD200 expression was widely detected in BCP-ALL; however, the expression pattern is variable and could be subtype specific, as shown in several studies [5,162,163]. Thus far, not much data on the CD200 levels are available for T-ALL.

### 7.2. Regulatory T Cells

Although the major function of regulatory T cells (Tregs) is maintaining self-tolerance [164], the accumulation and altered function of these cells has also been implicated in cancer progression [165]. In several tumor types, including hematological cancers, Tregs accumulation is associated with increased disease progression and suppression of anti-tumor responses [166,167,168,169]. The prevalence of the Tregs population in pediatric and adult ALL has been recently summarized by Niedzwiecki et al. [170]. In all the reviewed literature on ALL, the authors found increased frequency of Tregs in either peripheral blood or the BM of leukemic patients. Notably, rare studies provide a correlation of the Tregs number with disease progression or a detailed explanation of how they elicit potent suppressive mechanisms; thus, more functional studies in this context are of great need. Furthermore, in recent studies, an increased circulating Tregs frequency has been linked with diminished potency of selected immunotherapies against BCP-ALL, such as blinatumomab [6] or CAR-T cells [171]. A more detailed explanation of immunosuppression mediated by blinatumomab-stimulated Tregs in BCP-ALL involves not only IL-10 production but also direct cell-to-cell contacts with other T cell subtypes. Collectively, these results show that an elevated Tregs number is an important prognostic factor for poor response to some immunotherapies. Hence, strategies focused on Tregs ablation or selective inactivation as a component of combination therapies against ALL should be further exploited.

## 8. Conclusions

Studies conducted in the last decade revealed that ALL cells exploit various mechanisms to avoid immune recognition and destruction by the immune system. ALL cells escape from NK cell recognition and cytotoxicity by downregulating the ligands of the NK cell-activating receptors (MICA/B, ULBP1, and NEC-2) [119], resist phagocytosis by upregulating the CD47 “do not eat me signal” [105], and avoid a T cell response by triggering selected inhibitory checkpoints (TIM-3, CD200R) [5]. Moreover, accumulating evidence indicates that ALL-driven alterations in immune cells contribute to escape from immunosurveillance, leukemia progression, and thus affect treatment outcome. It has been recently demonstrated that the efficacy of chemo- and immunotherapies of ALL may be compromised by the accumulation of cells with immunosuppressive function, such as Tregs and non-classical monocytes [6,11,171]. However, more research is needed to assess the functional role of Tregs and to elucidate which properties of non-classical monocytes promote leukemic survival. Importantly, some strategies to target the leukemia-supporting interactions with the cells of the BM microenvironment already have been proposed in preclinical studies (Table 5).

The impairment of the numbers and function of effector immune cells (NK cells, T cells, and M1 macrophages) may diminish the efficacy of therapeutic modalities relying on the function of these effector cells. For example, functionally compromised and infrequent NK cells in ALL may negatively affect the outcome of treatment with therapeutic monoclonal antibodies. On the other hand, thanks to carefully optimized protocols of in vitro expansion and activation, healthy donor-derived NK cells have already demonstrated preclinical efficacy [125,126,127] and may pave the way for novel cell-based immunotherapies. However, these NK cell-based adoptive therapies must be further tested for their clinical efficacy.

Although some recent studies showed the occurrence of ALL neoantigen-specific T cells [3], further research is needed to elucidate if their numbers and activity is sufficient to employ strategies aimed at restoring T cell function with immune checkpoint blockade. Nevertheless, some clinical trials testing the efficacy of CTLA-4 and PD-1-mediated blockade have already been launched. Furthermore, the mechanisms behind T cell exhaustion in ALL are not yet fully understood. Preclinical studies indicate that TIM-3 rather than PD-1 or CTLA-4 is the major inhibitory checkpoint in ALL patients’ T cells [5,10]. Further studies are needed to show if TIM-3 expression affects the efficacy of ALL therapies and if TIM-3 can be a novel, useful immunotherapy target in ALL. These studies explaining the mechanisms of ALL T cells’ dysfunction may also contribute to improvements in CAR-based adoptive immunotherapy.

## Figures and Tables

**Figure 1 cancers-13-01536-f001:**
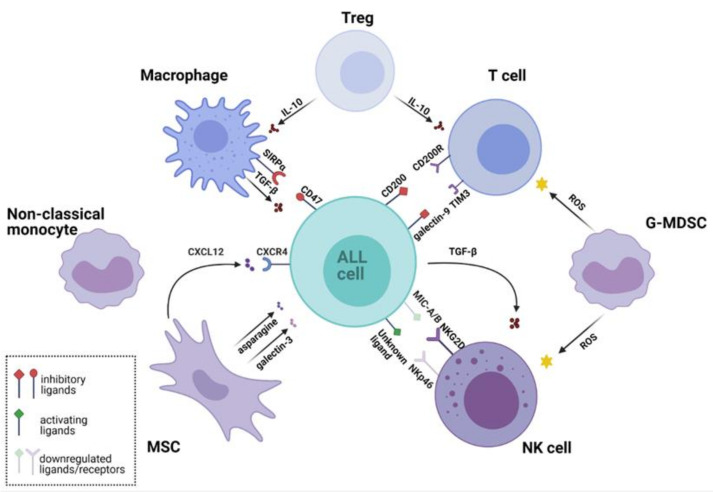
Leukemic microenvironment supports survival of acute lymphoblastic leukemia (ALL) cells and their immune evasion through multiple interactions. Various cell populations shape the leukemic microenvironment. Regulatory T cells (Tregs) secrete inhibitory cytokines that suppress the cytotoxic activity of T cells and reduce macrophage phagocytosis. Granulocytic Monocyte Derived Suppressor Cells (G-MDSC) produce reactive oxygen species (ROS) that inhibit activity of T cells and natural killer (NK) cells. NK cells express low levels of natural cytotoxicity triggering receptor p46 (NKp46) activating receptor, while ALL cells downregulate major histocompatibility complex class I-related chains A/B (MIC-A/B)-a ligand for natural killer group 2 member D (NKG2D) activating receptor. ALL also drives NK cell dysfunction by secreting immunosuppressive transforming growth factor beta (TGF-β). Mesenchymal stem cells (MSCs) secrete chemokines, e.g., C-X-C chemokine ligand 12 (CXCL12), which binds C-X-C chemokine receptor type 4 (CXCR4) and promotes ALL engraftment into the vascular niche. Furthermore, MSCs protect ALL cells against the treatment by secretion of galectin-3, which activates the nuclear factor kappa-light-chain-enhancer of activated B cells (NF-κB) pathway. MSCs also secrete metabolites, such as asparagine, which reduces the cytotoxicity of L-asparaginase. Non-classical (CD16^+^) monocytes infiltrate the leukemic microenvironment and are thought to be involved in ALL cells protection. Macrophages from the leukemic niche acquire immunosuppressive properties and secrete tumor-promoting cytokine TGF-β. Their phagocytic activity is reduced by the interaction of signal regulatory protein α (SIRPα) with cluster of differentiation 47 (CD47)—a “do not eat me signal” expressed by leukemic cells. The figure was created in BioRender (https://biorender.com/; accessed on 27 February 2021). Other abbreviations: CD200, cluster of differentiation 200; CD200R, CD 200 receptor; G-MDSC, Granulocytic Monocyte Derived Suppressor Cells; IL-10, Interleukin 10; TIM3, T-cell immunoglobulin domain and mucin domain 3.

**Table 1 cancers-13-01536-t001:** Molecular subtypes of B cell precursor acute lymphoblastic leukemia [1,2].

Molecular Subtype	Frequency	Prognosis	Specific Immunophenotypic Features
High hyperdiploid (>50 chromosomes)	25% children; 3% AYA and adults	Favorable	CD123pos [15], CD66c pos [16]
*ETV6-RUNX1* t(12;21)(p13;q22)	30% children; <5% AYA and adults	Favorable	CD66c neg, CD10pos, CD25neg, CD44 neg [17,18]
*DUX4* and *ERG*-deregulated ALL	5–10% acute lymphoblastic leukemia	Favorable	none
*TCF3-PBX1*, t(1;19) (q23;p13)	5% in children, rare in adults	Favorable	CD10pos, CD38pos, CD34 neg [18]
Internal amplification of chromosome 21 (iAMP21)	3% in children and AYA	Favorable with intensive therapy	none
*NUTM1*-rearranged	Exclusively in children (1%)	Favorable	NG2 pos [19]
*PAX5* alterations (fusion, mutation, amplification)	Highest in children (11%)	Intermediate	none
*ZNF384*-rearranged	5% children; 10% AYA and adults	Intermediate	CD10 weak/neg. CD13 pos, CD33 pos [20]
*PAX5* P80R	Highest in adults (4%)	Intermediate	none
*KMT2A*-rearranged	High in infants (90%) and adults (15%)	Poor	NG2 pos [21], CD38pos, CD10neg [18]
*BCR–ABL1*, Philadelphia chromosome [Ph], t(9;22) (q34;q11)	2–5% children, 6% AYA; >25% adults	Poor	CD66c pos, CD9 pos, CD 123pos, CD34 pos [18]
Philadelphia chromosome-like acute lymphoblastic leukemia	10% children; 25–30% AYA; 20% adults	Poor	TLSPR pos [22]
Low-hypodiploid (32–39 chromosomes)	10% adults; 5% AYA and >10% adults	Very poor	none
*BCL2/MYC* rearranged	3% AYA and adults	Poor	CD44 neg [23]
Near-haploid (24–31 chromosomes)	2% children; <1% AYA and adults	Poor	none
*MEF2D*-rearranged	4% children; 7% AYA and adults	Poor	CD10 neg CD38 pos [24]
*TCF3-HLF* t(17;19) (q22;p13)	<1% ALL	Poor	none
*ETV6-RUNX1*-like	3% in children	Unknown	none
*IKZF1* N159Y	<1% in all ages	Unknown	none

Abbreviations: AYA, adolescents and young adults; *BCL2*/*MYC*, BCL2 Apoptosis Regulator/MYC Proto-Oncogene; *DUX4*, Double Homeobox 4; *ETV6-RUNX1*, ETS Variant Transcription Factor 6—RUNX Family Transcription Factor 1; *ERG*, ETS Transcription Factor ERG; *IKZF1*, IKAROS Family Zinc Finger 1; *KMT2A*, Lysine Methyltransferase 2A; *MEF2D*, Myocyte Enhancer Factor 2D; NG2, Neural/glial antigen 2; *NUTM1*, NUT Midline Carcinoma Family Member 1; *PAX5*, Paired Box 5; TCF3-*PBX1*, Transcription Factor 3—Pre-B-Cell Leukemia Transcription Factor 1; *TCF3-HLF*, Transcription Factor 3—Hepatic Leukemia Factor; TLSPR, thymic stromal lymphopoietin receptor; *ZNF384*, Zinc Finger Protein 384.

**Table 2 cancers-13-01536-t002:** Molecular subtypes in T cell acute lymphoblastic leukemia, based on [28,29].

Subtype	Characteristics	Frequency	Dominant Genetic Abnormalities	Outcome
ETP (early T cell precursor)	Gene expression profile similar to hematopoietic stem cells and myeloid progenitors, with a high expression of self-renewal genes including *LMO2*/*LYL1 HOXA*, and *BCL2*	10%	Mutation of the JAK-STAT or Ras signaling pathways (e.g., *FLT3*, *NRAS* and *JAK3*), epigenetic regulators (e.g., *EZH2*, *IDH1*, *IDH2*, *DNMT3A*), genes involved in hematopoietic development (*GATA3*, *ETV6*, *RUNX1*, *IKZF1*), histone-modifying genes (*EZH2*, *EED*, *SUZ12*, *SETD2* and *EP300*), and rearrangements of *NUP98* and *KMT2A* genes	Can be effectively treated using early-response-based intensification
TLX3 (T-cell leukemia homeobox protein 3)	Lack a functional T-cell receptor (TCR) or presence of γ/δ TCR, rearrangements of the transcription factor *TLX3*	25%	High frequency of *NOTCH1* mutations, and *CDKN2A* deletions. Mutations in *CTCF*, *DNM2*, *PHF6*, *BCL11B*, *MYC*, *RPL5*, *RPL10*, *KDM6A* and *IL7R* genes	Favorable
TLX1/NKX2.1 (T-cell leukemia homeobox protein 1/NK2 homeobox 1)	Genomic rearrangements involving either *TLX1* or *NKX2.1*, CD1 expression, and differentiation arrest at the cortical stage, proliferative subtype	10%	*TLX1* or *NKX2.1* translocations	Excellent
TAL/LMO (transcription activator-like/LIM domain-only)	Ectopic expression of *TAL1*, *TAL2*, *LYL1*, *LMO1*, *LMO2*, or *LMO3* and late cortical immunophenotype	40–60%	Mutations of PI3K signaling pathway (*PTEN* and *PIK3R1*), *USP7* alterations, *LEF1* deletions, *SIL-TAL1* fusion	Poor

Abbreviations: BCL11B, BAF Chromatin Remodeling Complex Subunit; BCL2, BCL2 Apoptosis Regulator; CD1, cluster of differentiation 1; CDKN2A, Cyclin Dependent Kinase Inhibitor 2A; CTCF, CCCTC-Binding Factor; DNM2, Dynamin 2; DNMT3A, DNA Methyltransferase 3 Alpha; EED, Embryonic Ectoderm Development; EP300, Histone Acetyltransferase P300; ETV6, ETS Variant Transcription Factor 6; EZH2, Enhancer Of Zeste 2 Polycomb Repressive Complex 2 Subunit; FLT3, Fms Related Receptor Tyrosine Kinase 3; GATA3, GATA Binding Protein 3; HOXA, Homeobox A Cluster; IDH1, Isocitrate Dehydrogenase (NADP(+)) 1; IDH2, Isocitrate Dehydrogenase (NADP(+)) 2; IKZF1, IKAROS Family Zinc Finger 1; IL7R, Interleukin 7 Receptor; JAK3, Janus Kinase 3; JAK-STAT, Janus kinase (JAK)-signal transducer and activator of transcription (STAT) pathway; KDM6A, Lysine Demethylase 6A; KMT2A, Lysine Methyltransferase 2A; LEF1, Lymphoid Enhancer Binding Factor 1; LMO1, LIM Domain Only 1; LMO2/LYL1, LIM Domain Only 2/ Lymphoblastic Leukemia Associated Hematopoiesis Regulator 1; LMO3, LIM Domain Only 3; MYC, MYC Proto-Oncogene; NOTCH1, Notch Receptor 1; NRAS, Neuroblastoma RAS Viral Oncogene Homolog; NUP98, Nucleoporin 98 And 96 Precursor; PHF6, PHD Finger Protein 6; PI3K, Phosphoinositide 3-kinase; PIK3R1, Phosphoinositide-3-Kinase Regulatory Subunit 1; PTEN, Phosphatase And Tensin Homolog; Ras, Ras Family Small GTP Binding Protein; RPL10, Ribosomal Protein L10; RPL5, Ribosomal Protein L5; RUNX1, RUNX Family Transcription Factor 1; SETD2, SET Domain Containing 2; SIL-TAL1, STIL Centriolar Assembly Protein- TAL BHLH Transcription Factor 1 gene fusion; SUZ12, Suppressor Of Zeste 12 Protein Homolog; TAL1, TAL BHLH Transcription Factor 1; TAL2, TAL BHLH Transcription Factor 2; USP7, Ubiquitin Specific Peptidase 7.

**Table 4 cancers-13-01536-t004:** Overview of the inhibitory immune checkpoint interactions.

Checkpoint Molecule	Cell Source	Ligand	Ligand Cell Source	Main Mechanism of Action	Selected Literature
CTLA-4	Activated CD4/CD8^+^ T cells, Tregs, some cancer cells	CD80/CD86	APCs	outcompetes CD28 for interaction with CD80/CD86 and blocks T cells activation	[143,144,145]
PD-1	Activated CD4/CD8^+^ T cells in the periphery, activated DCs, B cells, NK cells	PDL-1/2	APCs, T cells, non-lymphoid tissues, several tumor types	inhibits T cells expansion and their effector functions	[146,147,148,149]
TIM-3	CD4/CD8^+^ T cells, Tregs, NK cells, myeloid cells	Gal-9, HMGB1, PtdSer, Ceacam-1	Endothelial, haematopoietic cells, several tumor types	Gal-9 binding disrupts formation of immune synapse and leads to apoptosis; HMGB1 binding inhibits DCs function	[150,151,152]
LAG-3	Activated CD4/CD8^+^ T cells, Tregs, NK and NKT cells	MHC II	APCs	Influences on the proliferation and cytokine production of T cells; LAG-3 expression increases IL-10 production by Tregs	[152,153]
TIGIT	Activated CD4/CD8^+^ T cells, Tregs, NK cells	CD155	APCs, activated T cells, several tumor types	directly inhibits T cells and NK cells; induces tolerogenic DCs; stimulates Tregs function	[154]
CD200R	APCs, myeloid cells, CD4/CD8^+^ T cells, Tregs	CD200	Activated T cells, B cells, several tumor types, non-lymphoid tissues	inhibits production of IL-2 and IFN-γ by macrophages; inhibits phagocytosis; inhibits NK cells function; unclear mechanism of T cells suppression	[155,156,157]

Abbreviations: CTLA-4, cytotoxic T-lymphocyte-associated protein 4; APCs, antigen presenting cells; PD-1, programmed death receptor 1; PD-L1/2, programmed death-ligand 1/2; TIM-3, T cell immunoglobulin and mucin-domain containing-3; Gal-9, Galectin-9; HMGB1, high mobility group protein B1; PtdSer, phosphatidylserine; Ceacam-1, carcinoembryonic antigen cell adhesion molecule 1; LAG-3, lymphocyte activation gene 3; TIGIT, T-cell immunoglobulin and ITIM domain; CD200R, CD200 receptor.

**Table 5 cancers-13-01536-t005:** Overcoming immune evasion and targeting leukemia-promoting interactions provided by the cells of the bone marrow niche—preclinical strategies.

Leukemia-Promoting Mechanism	Treatment Strategy	Observed Effects	Models	Literature
Overexpression of adhesion molecules, Stem cell-like phenotype in *IKZF1*-mutated BCP-ALL	Retinoids FAK inhibitors	Abrogation of adhesion and self-renewal Increased sensitivity to dasatinib	In vitro murine and human BCP-ALL cells In vivo, murine model of Ph-positive BCP-ALL	[70]
Accumulation of leukemia-promoting myeloid cells	CSF1R blockade	Depletion of myeloid cells Increased sensitivity to nilotinib	In vivo, murine model of Ph-positive BCP-ALL	[11]
	Clodronate in liposomes	Depletion of myeloid cells Diminished leukemia burden Prolonged survival	In vivo, LN3 T-ALL transgenic murine model	[104]
Overexpression of CD47 anti-phagocytic protein by BCP- and T-ALL cells	Antibodies anti-CD47	Increased phagocytosis Inhibition of leukemia engraftment	In vitro phagocytosis assay In vivo, PDX model of BCP- and T-ALL	[105]
Insufficient T cell-dependent immune response	Monocytes engineered to express IFNα	Promotion of T cell activity Improvement of ICI and CAR T cell immunotherapy	In vivo, murine model of BCP-ALL transplanted with monocytes expressing IFNα	[106]
Leukemia-driven T cell dysfunction	T cells isolated from leukemic mice, modified with CAR	Partial leukemia eradication	In vivo, murine model of *TCF3/PBX1* BCP-ALL	[159]
	T cells isolated from non-leukemic mice, modified with CAR	Complete leukemia eradication	In vivo, murine model of *TCF3/PBX1* BCP-ALL	[159]

Abbreviations: BCP-ALL, B cell precursor acute lymphoblastic leukemia; CAR, chimeric antigen receptor; CSF1R, colony stimulating factor 1 receptor 1; FAK, focal adhesion kinase; IFNα, interferon α; IKZF1, IKAROS Family Zinc Finger 1; Ph-positive, Philadelphia-positive; T-ALL, T cell acute lymphoblastic leukemia; TCF3/PBX1, Transcription Factor 3/Pre-B-Cell Leukemia Homeobox 1 fusion protein.

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
