# Peer review of "Mechanisms of Immune Evasion in Acute Lymphoblastic Leukemia"

_cancers, 2021, doi:10.3390/cancers13071536_

Round 1

Reviewer 1 Report

Manuscript „Advances in understanding and targeting tumor-supporting interactions between acute lymphoblastic leukemia and the bone marrow niche“ by Agata Pastorczak and co-workers.

In this manuscript, the authors focus on acute lymphoblastic leukemia (ALL) and recent advances with respect to its interaction with the bone marrow microenvironment for both, leukemogenesis and therapeutic resistance. It is a comprehensive review giving detailed information on several aspects of this highly relevant topic.

Comment:

  • When providing a stae-of-the-art review on ALL, the current WHO classification should be stated and discussed (Arber et al. Blood 2016)
  • Although HSCT is mentioned as the first „immunotherapy“, it is described in a very abbreviated way with mainly its side effects being mentioned. However, it might be worth mentioning that intial observations of the association of cGVHD with reduced relapse rates formed the basis for that assumption. And for some ALL subtypes, this procedure significantly increases the curative potential.
  • A current oncogenic concept comprise tumor heterogeneity and its role for clonal evolution and therapeutic resistance. This should also be discussed for ALL, also in light of novel „immunotherapies“.
  • Similar to AML, ALL primarily transforms a hitherto normal microenvironment. This statement should be corroborated by quoting respective reports, preferably on the (normal) genetics of the microenvironment in that disorder.
  • Tables 1 and 2 need a quotation in their legends.
  • Figure 1 should be shown initially in the manuscript.

Author Response

We thank the Reviewer for all the valuable comments. Point-by-point responses are below: 

  1. When providing a state-of-the-art review on ALL, the current WHO classification should be stated and discussed (Arber et al. Blood 2016).

In the Chapter 2 we now refer to the current WHO classification. We also added the following text to the Paragraph 2:

“This heterogeneous mutational landscape of BCP-ALL is reflected in the revised version of World Health Organization (WHO) classification of myeloid neoplasms and acute leukemias which distinguishes eleven subtypes of ALL, based on the presence of somatic molecular lesions.”

  1. Although HSCT is mentioned as the first „immunotherapy“, it is described in a very abbreviated way with mainly its side effects being mentioned. However, it might be worth mentioning that intial observations of the association of cGVHD with reduced relapse rates formed the basis for that assumption. And for some ALL subtypes, this procedure significantly increases the curative potential.

We thank reviewer for this remark, in the revised version of the manuscript we added the information about the beneficial role of HSCT in patients with ALL and we provide the references that discuss this topic extensively from adult ALL and pediatric ALL perspectives.

We added the following text to the Paragraph 3:

„The curative effect of HSCT results from direct cytotoxicity from the chemo-radiotherapy administered in the conditioning regimen, along with an immune effect termed graft-versus-leukemia (GVL)[31].”

“Whereas HSCT is still perceived as a standard consolidative therapy preventing relapse in many adult patients with Ph-negative ALL [32], the role of HSCT in childhood ALL is continuously redefined as advances in immunotherapy and targeted treatment are made. Detailed discussion of indications to HSCT in childhood ALL depending on molecular aberrations and response to therapy was presented by Merli  et al [33].”

  1. A current oncogenic concept comprise tumor heterogeneity and its role for clonal evolution and therapeutic resistance. This should also be discussed for ALL, also in light of novel „immunotherapies“.

The topic of clonal evolution and resistance to immunotherapy is important but complex and not fully understood. Therefore, in this review we decided to shortly explain the significant role of clonal evolution in response to treatment, illustrate it by an example and provide appropriate references which extensively discuss this issue. We added the following text at the end of the Paragraph 3:

„The long-term response to immune-based therapy might be limited as a result of clonal evolution of leukemia. This process promotes selection of ALL subclones exhibiting treatment-refractory phenotype through progressive and diverse accumulation of genetic alterations by neoplastic cells under therapeutic pressure [54,55]. There are several examples proving that clonal evolution and intratumor heterogeneity affect the efficacy of targeted immunotherapy [56,57]. One of the most illustrative in this context is successful elimination of CD19+ cells by patients with MLL-rearranged B-ALL receiving CD19 CAR-T [54].”

  1. Similar to AML, ALL primarily transforms a hitherto normal microenvironment. This statement should be corroborated by quoting respective reports, preferably on the (normal) genetics of the microenvironment in that disorder.

We provided the information about the presence of genetic alterations and transcriptomic variations in the stromal compartment at diagnosis of leukemia (Chapter 4.1):

‘’ Leukemogenesis can no longer be perceived as a process that exclusively affects HSC or      more differentiated lineage precursors. Some genetic alterations, epigenetic changes and transcriptomic variations are already present within stromal compartment at cancer diagnosis, but their incidence in patients with ALL has not been precisely defined [62-64]. Moreover, the possible role of microenvironment in promoting hematopoietic dysregulation is not restricted to the small percentage of patients harboring germline predisposition to leukemia. At least, in case of myeloid malignancies genomic and transcriptomic variations in stromal compartment have been also observed outside of constitutional defects [65].’’

  1. Tables 1 and 2 need a quotation in their legends.

We added references to Table 1 and Table 2 accordingly.

  1. Figure 1 should be shown initially in the manuscript.

The Figure 1 is now moved to the Introduction.

Reviewer 2 Report

In this manuscript, Pastorczak et al. extensively reviewed the genotype & immunophenotype, chemo- & immunotherapy, crosstalk between cancer cell and bone marrow niche (especially mesenchymal stem cells and other immune cells) of acute lymphoblastic leukemia (ALL), providing a comprehensive summary of both understanding and targeting of ALL. Although the manuscript is well presented in general, the reviewer has a concern about the abstract.

Based on the structure of abstract, most of the sentences are introduction, while only last two sentence describe the content of review. Authors should give a brief introduction and provide more detailed information about the content of the manuscript. The abstract states that “Interactions between ALL cells and the BM microenvironment protect cancer cells against chemotherapy-induced death, promote their survival and contribute to disease relapse” and “Importantly, these alterations in immune cells also affect the efficacy of both chemo- and immunotherapy”, while these statement lack support from the text. Please consider rewriting the abstract.

Author Response

We thank the Reviewer for all the valuable comments. Point-by-point responses are below: 

In this manuscript, Pastorczak et al. extensively reviewed the genotype & immunophenotype, chemo- & immunotherapy, crosstalk between cancer cell and bone marrow niche (especially mesenchymal stem cells and other immune cells) of acute lymphoblastic leukemia (ALL), providing a comprehensive summary of both understanding and targeting of ALL. Although the manuscript is well presented in general, the reviewer has a concern about the abstract.

Based on the structure of abstract, most of the sentences are introduction, while only last two sentence describe the content of review. Authors should give a brief introduction and provide more detailed information about the content of the manuscript. The abstract states that “Interactions between ALL cells and the BM microenvironment protect cancer cells against chemotherapy-induced death, promote their survival and contribute to disease relapse” and “Importantly, these alterations in immune cells also affect the efficacy of both chemo- and immunotherapy”, while these statement lack support from the text. Please consider rewriting the abstract.

We thank the reviewer for this remark. We rewrote the abstract according to reviewer’s suggestions, and now it is as follows:

„Acute lymphoblastic leukemia (ALL) results from a clonal expansion of abnormal lymphoid progenitors of B cell (BCP-ALL) or T cell (T-ALL) origin that invade bone marrow, peripheral blood, and extramedullary sites. Leukemic cells, apart from oncogenes-driven ability to proliferate and avoid differentiation, also change the phenotype and function of innate and adaptive immune cells, leading to escape from the immune surveillance. In this review, we provide an overview of the genetic heterogeneity and treatment of BCP- and T-ALL. We outline interactions of leukemic cells in the bone marrow microenvironment, mainly with mesenchymal stem cells and immune cells. We describe the mechanisms by which ALL cells escape from immune recognition and elimination by the immune system. We focus on the alterations in ALL cells, such as overexpression of ligands for various inhibitory receptors, including anti-phagocytic receptors on macrophages, NK cell inhibitory receptors, as well as T cell immune checkpoints. In addition, we describe how developing leukemia shapes the bone marrow microenvironment and alters the function of immune cells. Finally, we emphasize that immunosuppressive microenvironment can reduce the efficacy of chemo- and immunotherapy and provide examples of preclinical studies showing strategies for improving ALL treatment by targeting these immunosuppressive interactions.”

Reviewer 3 Report

  1. This is a review about t he bone marrow niche, but looking at the subheaders, there is no focus and much information on this subject.
  2. The title mentions”targeting”, no table , Figure or subheader refers to this. One table lists immunotherapeutic strategies, but they do not target the interaction with the BM niche.
  3. Table 1- right column will need reference for “immunophenotypic feature”. Suggest also to add more than one marker if possible.
  4. Figure 1 – incomplete, please check if all cellular components mentioned in this review are included, e.g. dendritic cells are missing.
  5. This is a review on leukemia, taking up 1/3 of the text or more. The “advances in understanding of… BM niche interactions” are not clear and not well described.

Author Response

We thank the Reviewer for all the valuable comments. The point-by-point answers are provided below:

  1. This is a review about the bone marrow niche, but looking at the subheaders, there is no focus and much information on this subject.

We agree that the title and the abstract of the previous version of the manuscript were misleading. Considering this remark of the Reviewer and the fact that we mainly focus on interactions between leukemic and immune cells, we decided to change the title of the manuscript to the one which better reflects the content of the article. The new title is as follows: “Mechanisms of Immune Evasion of Acute Lymphoblastic Leukemia”.

  1. The title mentions”targeting”, no table , Figure or subheader refers to this. One table lists immunotherapeutic strategies, but they do not target the interaction with the BM niche.

We added an additional table (Table 5) which specifically refers to the mechanisms of immune evasion of ALL and to therapeutic strategies that can overcome them. We also changed the tile of the manuscript.

  1. Table 1- right column will need reference for “immunophenotypic feature”. Suggest also to add more than one marker if possible.

We added references to Table 1 and when it was possible, we extended the list of markers specific for particular genetic subtypes.

  1. Figure 1 – incomplete, please check if all cellular components mentioned in this review are included, e.g. dendritic cells are missing.

We agree with the Reviewer that the Figure 1 is incomplete. We selected only the most important interactions between leukemic and immune cells for graphical representation in order to make Figure 1 legible and informative for a reader. The crosstalk between dendritic cells and ALL cells is now still incompletely understood.

  1. This is a review on leukemia, taking up 1/3 of the text or more. The “advances in understanding of… BM niche interactions” are not clear and not well described.

We believe that changing the misleading title and abstract as well as adding Table 5 should now have a positive effect on consistency with the content of the manuscript. However, we agree with the Reviewer that we provide a very detailed introduction to ALL genetic heterogeneity (Paragraph 2, Table 1, 2). We believe that this may be useful for some readers, but if the Reviewer and Editors believe otherwise, we can delete this part of text and tables and provide only appropriate references instead.

Round 2

Reviewer 3 Report

No  further  comments